# Unraveling the mechanisms of PAMless DNA interrogation by SpRY-Cas9

Grace N. Hibshman [1,2], Jack P. K. Bravo[1,5] ✉, Matthew M. Hooper [1,2], Tyler L. Dangerfield[1], Hongshan Zhang[1,3], Ilya J. Finkelstein [1,3], Kenneth A. Johnson[1,2] & David W. Taylor [1,2,3,4] ✉

CRISPR-Cas9 is a powerful tool for genome editing, but the strict requirement for an NGG protospacer-adjacent motif (PAM) sequence immediately next to the DNA target limits the number of editable genes. Recently developed Cas9 variants have been engineered with relaxed PAM requirements, including SpG-Cas9 (SpG) and the nearly PAM-less SpRY-Cas9 (SpRY). However, the molecular mechanisms of how SpRY recognizes all potential PAM sequences remains unclear. Here, we combine structural and biochemical approaches to determine how SpRY interrogates DNA and recognizes target sites. Divergent PAM sequences can be accommodated through conformational flexibility within the PAM-interacting region, which facilitates tight binding to off-target DNA sequences. Nuclease activation occurs ~1000-fold slower than for *Streptococcus pyogenes* Cas9, enabling us to directly visualize multiple on-pathway intermediate states. Experiments with SpG position it as an intermediate enzyme between Cas9 and SpRY. Our findings shed light on the molecular mechanisms of PAMless genome editing.

CRISPR-Cas9 (clustered regularly interspaced short palindromic repeats, CRISPR-associated) is an RNA-guided endonuclease that uses RNA-DNA complementarity to precisely target and cleave double-stranded DNA[1]. To find its target, Cas9 must meticulously search double-stranded DNA across the entire genome to identify regions of gRNA (guide RNA) complementarity[2,3]. This search is simplified via the protospacer-adjacent motif (PAM) located immediately next to the targeted sequence[4,5]. The PAM acts as a rapid and efficient initial filter for possible target sites, circumventing the slow, kinetically unfavorable process of initiating duplex melting and R-loop formation. In the case of *Streptococcus pyogenes* Cas9 (henceforth referred to as Cas9), the required PAM sequence is NGG (where N can represent any nucleotide and G signifies guanine). In its native context, the PAM also serves as a mechanism to distinguish between self and non-self sequences[6–8].

For genome editing applications, the PAM requirement imposes a major limitation on the number of sequences that can be targeted[9].

Protein engineering efforts have focused on re-targeting or reducing the PAM specificity[10–15]. A PAMless Cas9 variant, termed SpRY, is different in that it is capable of targeting nearly any site in vivo[16]. However, since PAM recognition is a prerequisite for efficient DNA melting and R-loop formation, it is unclear how SpRY can identify diverse target sequences.

Here, we show that SpRY binds diverse PAM sequences by forming non-specific interactions with the phosphodiester backbone of the target DNA. The non-specific interactions employed by SpRY causes the enzyme to accumulate at off-target sites where DNA melting has initiated, but the R-loop cannot complete due to the lack of complementarity. The reduced rate of DNA melting by SpRY activity enabled us to capture R-loop formation intermediates in real-time using kinetics-guided cryo-EM. Our combinatorial approach of kinetic characterization, structural elucidation, and single-molecule imaging reveals the molecular

[1]Department of Molecular Biosciences, University of Texas at Austin, Austin, TX 78712, USA. [2]Interdisciplinary Life Sciences Graduate Programs, Austin, TX 78712, USA. [3]Center for Systems and Synthetic Biology, University of Texas at Austin, Austin, TX 78712, USA. [4]LIVESTRONG Cancer Institutes, Dell Medical School, Austin, TX 78712, USA. [5]Present address: Institute of Science and Technology Austria (ISTA), Klosterneuburg, Austria. ✉e-mail: jack.bravo@ist.ac.at; dtaylor@utexas.edu

mechanisms of PAMless DNA targeting by SpRY, informing its use as a tool in biotechnology.

## Results

### Promiscuous PAM recognition by SpRY

The Cas9 residues R1333 and R1335 confer specificity for two guanines in the second and third positions of the PAM by forming four hydrogen bonds with the Hoogsteen faces[6]. Both residues are mutated in SpRY (R1333P, R1335Q) along with nine additional mutations (A61R, L1111R, D1135L, S1136W, G1218K, E1219Q, N1317R, A1322R, and T1337R) that facilitate PAMless DNA targeting.

To understand how SpRY can recognize diverse PAM sequences, we determined the structures of SpRY bound to NGG, NAC, and NTC PAM-containing DNA substrates with global resolutions of 3.3, 2.8 and 3.7 Å, respectively (Fig. 1 and Supplementary Fig. 1, Supplementary Table 1). All three structures were in the product state, with the HNH endonuclease domain repositioned at the cleaved scissile phosphate of the target strand (TS).

Comparison of these atomic models with a previously determined structure of Cas9 in the product state (PDB 7S4X[17]) shows that most of the complex is unaltered (RMSD of <1 Å), except for the PAM-interacting (PI) domain. While R1335Q makes hydrogen bond contacts with the PAM bases, the mutated residues L111R, G1218K, E1219Q, and T1337R make several electrostatic interactions with the target DNA duplex and increase the overall net charge of the PI domain relative to Cas9 (Fig. 1d–g). This change in protein surface potential may stimulate the formation of non-specific electrostatic interactions with DNA to compensate for the loss of specific hydrogen bond contacts with the PAM nucleobases, enabling DNA interrogation.

Structures of the PAM-reprogrammed Cas9 variants VQR and VRQR exhibit a slight shift in the PAM region of the duplex backbone by ~1–2 Å[18,19]. We observed a similar phenomenon in SpRY where the DNA backbone is shifted by ~5 Å relative to the equivalent position for Cas9 (Supplementary Fig. 13). This shift in the DNA backbone is mediated by the bulky, hydrophobic mutations D1135L and S1136W wedged into the minor groove.

In addition to the SpRY mutations that directly engage with the PAM, three more mutations surround the PAM readout region. N1317R and A1322R form electrostatic interactions with the phosphate backbone of the displaced NTS immediately adjacent to the PAM (i.e., positions 1 and 2). These interactions likely favor initial melting of the DNA duplex. Additionally, A61R forms ionic interactions with E1108 at the base of the bridge helix and U63 of the gRNA scaffold, directly behind the first base of the R-loop, potentially supporting R-loop initiation (Supplementary Fig. 2).

Structures of SpRY in complex with NGG, NAC, and NTC PAM sequences are largely similar. Surprisingly, G1218K, N1317R, A1322R, and T1337R adopt different rotamers in the structures presumably to maximize the energetics of binding through the formation of non-specific interactions compatible with the different PAM sequences (Fig. 1h–j and Supplementary Fig. 2). This is in stark contrast to Cas9 in which R1333 and R1335 are pre-ordered and poised to quickly recognize the NGG PAM. We propose that in the absence of DNA targets, these solvent-exposed residues are highly flexible and can therefore adopt different conformations to accommodate diverse PAM sequences upon binding DNA. This observation suggests that SpRY recognizes DNA targets predominantly via non-specific electrostatic contacts rather than base-specific contacts.

### Single-molecule visualization of DNA interrogation by SpRY

Given the divergent PAM recognition mechanism of SpRY, we wanted to directly visualize DNA target search on long DNA molecules (Lambda DNA) in comparison to Cas9[20] (Fig. 2). To do so, we used single-molecule DNA curtains with dSpRY and dCas9 fluorescently labeled with anti-FLAG coupled quantum dots, as previously

described[21]. In this way, we can image SpRY probing diverse sequence space to locate a target[20,21]. We first injected dCas9 gRNP into single-tethered DNA curtains, and then flushed out excess protein with high salt buffer. As expected, dCas9 gRNPs preferentially bound to the target site[22] (Fig. 2a–c). In contrast, on the timescale of the experiment, dSpRY gRNPs bound non-specifically along the entire length of the DNA with no significant enrichment at the target site.

We next used double-tethered DNA curtains to visualize transient interactions with off-targets throughout the long DNA molecule. In these assays, both DNA ends are affixed to microfabricated barriers, allowing direct observation of protein binding and dissociation along the length of the DNA without buffer flow (Fig. 2d, e). This enables us to record the locations and corresponding lifetimes of all binding events.

When bound to the target DNA sequence, the lifetime distributions for dSpRY and dCas9 were nearly identical ($t_{1/2}$ of $360 \pm 10$ s and $360 \pm 5$ s, respectively) (Fig. 2f). In contrast, dCas9 had a shorter binding lifetime on off-target DNA, and dSpRY remained associated with nearly the same lifetime as at the on-target sequence (Fig. 2g). These results demonstrate that SpRY remains stably bound to off-target sequences, taking longer to locate the correct target than Cas9. This is independently corroborated through in vivo single-particle tracking of SpRY and Cas9[23].

### SpRY tightly associates with off-target DNA

Our observation that SpRY binds DNA indiscriminately led us to question the binding affinity of SpRY to on- and off-target DNA sequences. Due to the spatial and temporal limitations of DNA curtains, we chose to use short (55 bp) DNA substrates for all subsequent experiments. This way we can tightly control our measurements to reveal the underlying molecular mechanisms of PAMless DNA interrogation by SpRY.

To further explore the capacity of SpRY to stably bind to off-target sequences, we measured the apparent binding affinity ($K_{d,app}$) of SpRY and Cas9 to an NGG PAM DNA substrate with a scrambled protospacer that lacked complementarity to the gRNA. As anticipated, Cas9 showed no significant binding within the concentration regime measured, up to 1000 nM, putting a lower limit of 8 μM on the $K_d$ for this DNA substrate (Fig. 3a). This measurement agrees with previous reports where Cas9 is weakly bound to PAM sequences with an apparent affinity >10 μM[24]. Strikingly, SpRY tightly bound to this substrate with a $K_{d,app}$ of $6 \pm 0.5$ nM.

We then prepared cryo-EM samples of SpRY in complex with the same off-target DNA substrate to visualize how SpRY recognizes this scrambled substrate. It was clear during classification and refinement that this dataset contained conformationally heterogeneous structures, which we separated through 3D classification. Within this ensemble, we could resolve four distinct structures, corresponding to 1, 6, 8, and 10 bp of R-loop formation, where the 6, 8, and 10 bp structures contain mismatches (Fig. 3b and Supplementary Fig. 4).

Serendipitously, the scrambled DNA sequence contained a four nucleotide stretch of complementarity to the gRNA, located in the NTS outside of the typical protospacer region (GCGT). The mismatch-containing 6, 8, and 10 bp structures align with this region of the DNA sequence where the first four bases of the R-loop are complementary to the gRNA with subsequent mismatches (Fig. 3c). As observed for Cas9[23], we observe REC2 and REC3 movement in the 6, 8, and 10 bp structures as the R-loop propagates past that structural checkpoint. This phenomenon supports a model where the REC2 and REC3 domains must be displaced during R-loop formation to accommodate the propagating gRNA:TS duplex.

Our mismatch-bound structures portray an ensemble of possible conformational states that SpRY can adopt during target search. These structures provide insight into what conformations SpRY may be trapped in during target search as the phage Lambda DNA used for

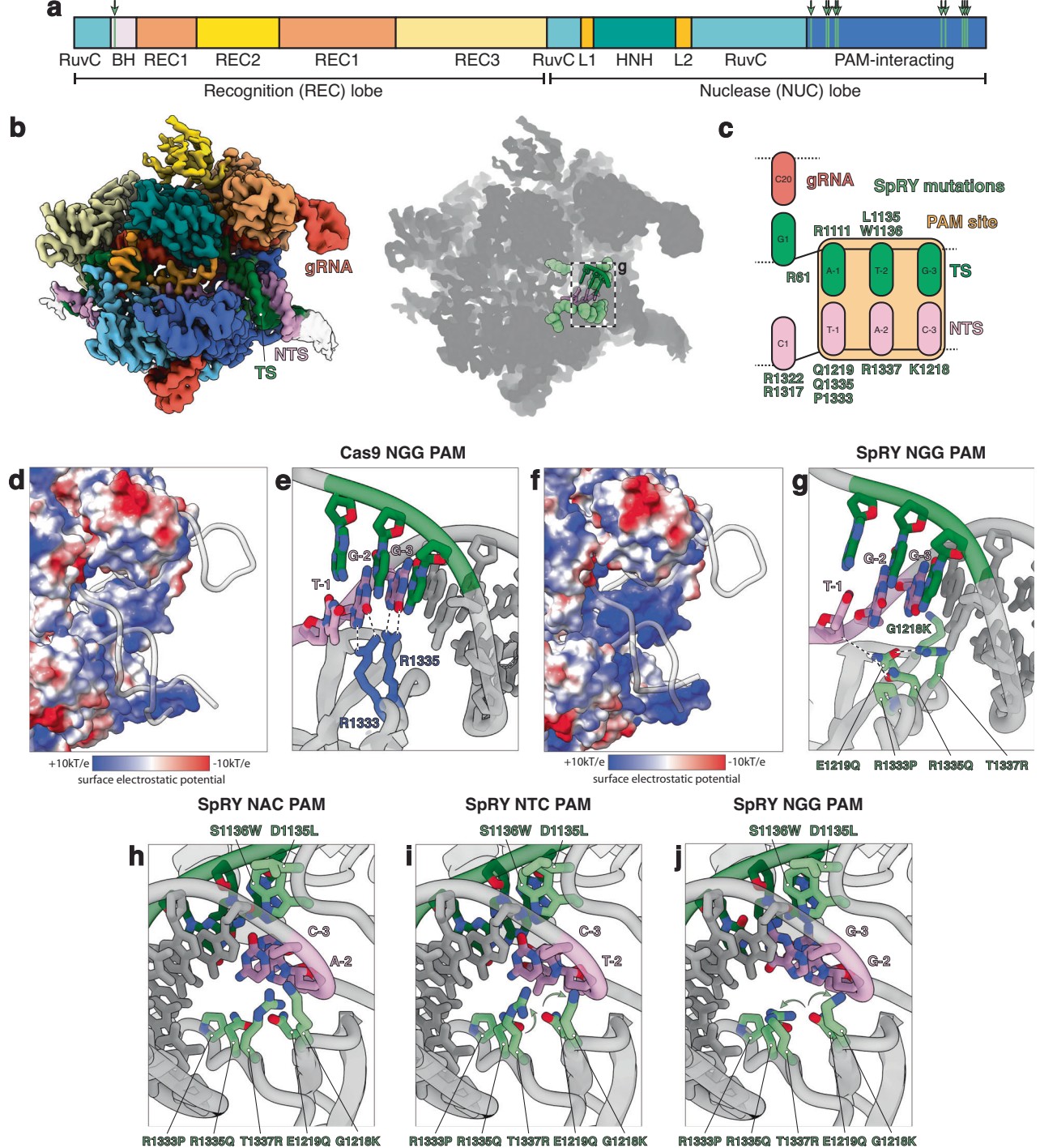

**Fig. 1 | Structural basis of PAMless DNA recognition. a** Cas9 domain organization with SpRY mutations A61R, L1111R, D1135L, S1136W, G1218K, E1219Q, N1317R, A1322R, R1333P, R1335Q, and T1337R highlighted in light green. **b** 2.8 Å cryo-EM reconstruction of SpRY bound to target DNA with TAC PAM sequence in the product state. Shadow of the SpRY NAC PAM product state structure with the PAM nucleotides shown as sticks and SpRY amino acid mutations shown as spheres. Molecules are colored: TS, green; NTS, pink; gRNA, red; SpRY mutations, light green. **c** Schematic representation of the SpRY PAM site interaction. The PAM site is highlighted in gold. **d** Surface electrostatic potential map of Cas9. **e** Detailed Cas9 NGG PAM site interaction (PDB: 7s4x). **f** Surface electrostatic potential map of SpRY. **g** Detailed SpRY NGG PAM site interaction. **h** View of NAC, **i** NTC, and **j** NGG PAM site interaction turned 180° relative to (**g**). Source data are provided as a Source Data file.

DNA curtains experiments contains almost 200 "GCGT" motifs evenly dispersed throughout the entire sequence.

Without PAM specificity, SpRY locates partial target sequences from which it cannot readily dissociate, nor achieve nuclease activation. Slow dissociation from non-productive states may create a significant block during target search by PAMless variants. We therefore hypothesized that our other SpRY cryo-EM datasets may contain structures of SpRY bound to off-target regions of the DNA substrate.

We revisited our largest cryo-EM dataset of SpRY in complex with NAC PAM DNA to search for any minor species that may represent non-productive complexes. Surprisingly, when we re-performed 2D classification using a larger particle box size, we observed a subset of

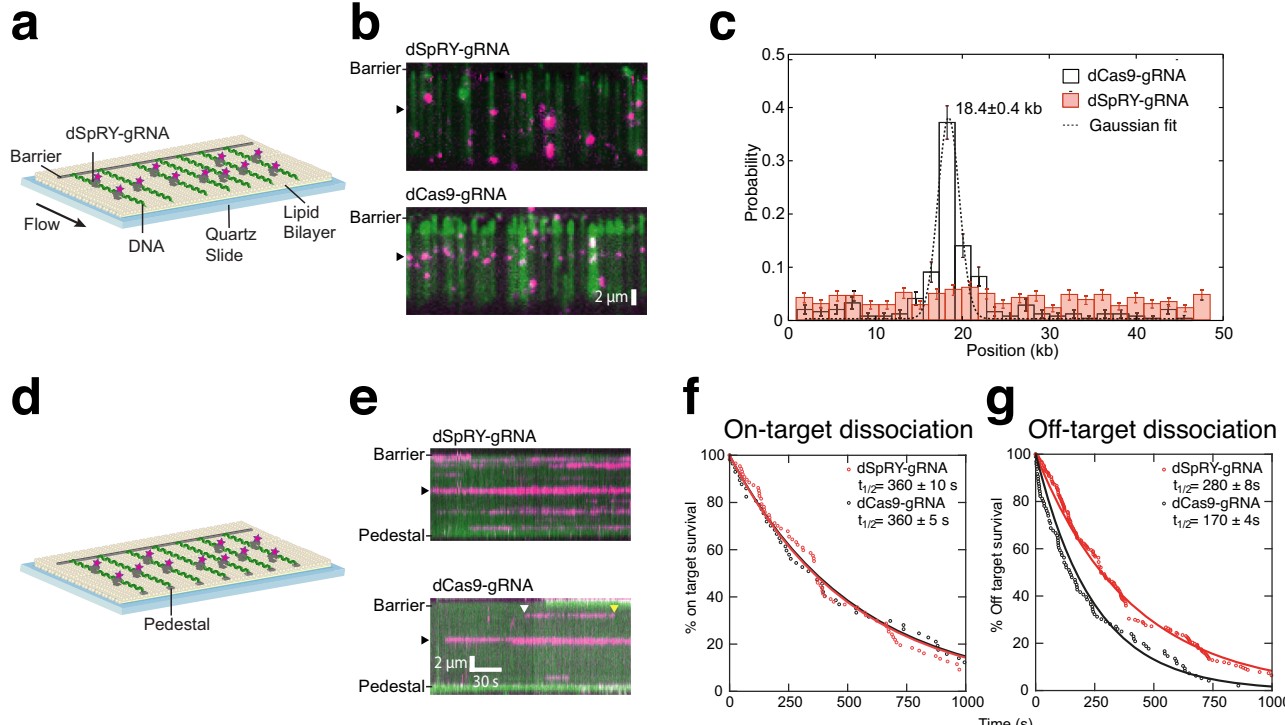

**Fig. 2 | SpRY accumulates at off-target sequences during target search.**
**a** Schematic of a single-tethered DNA curtain. **b** YOYO-stained DNA (green) bound by fluorescently labeled dSpRY-gRNA (top, magenta) or dCas9-gRNA (bottom, magenta). **c** Binding distribution of dSpRY-gRNA (*n* = 543) and dCas9-gRNA (*n* = 242); error bars represent 95% confidence intervals and measure of center is mean observations. **d** Schematic of a double-tethered DNA curtain.

**e** Kymographs depicting distinct binding events for dSpRY-gRNA and dCas9-gRNA. The white arrow and yellow arrow indicates the binding and releasing time point of dCas9-gRNA at the off-target site, respectively. **f** Lifetime of dSpRY-gRNA and dCas9-gRNA on target and (**g**) off-target. The half-lives for each enzyme are indicated. Source data are provided as a Source Data file.

particles that were much larger than anticipated. 3D reconstruction of these particles revealed the presence of a dimeric SpRY complex, with two SpRY gRNPs bound to the same DNA substrate facing opposite directions—we refer to this structure as a "SpRYmer" (Fig. 3d). Within the SpRYmer structure, the first SpRY is in the product state, with both TS and NTS cleaved. The second SpRY is bound to the same DNA molecule through its PI domain but has not initiated DNA unwinding likely due to the lack of substrate complementarity. Dimeric Cas9 has been observed during target search when two PAM sites are proximally located[25]. SpRYmer assembly is stabilized by direct DNA linkage, and by a kissing loop interaction formed between the nexus stem loop of each gRNP which was not observed in the Cas9 dimer (Fig. 3e and Supplementary Fig. 3). This unanticipated structure further supports the propensity of SpRY to stably bind to DNA irrespective of sequence. Notably, similar SpRYmer 2D classes were observed for NGG and NTC PAM datasets, but such classes were poorly resolved due to low abundance.

We propose that high affinity binding to off-target DNA sequences is a direct consequence of the SpRY mutations which create a large, positively charged patch within the PI domain. This patch acts as an anchor for DNA binding, facilitating stable non-specific electrostatic interactions with duplexes prior to unwinding. This important property of SpRY compensates for the loss of low affinity, rapid PAM probing employed by Cas9.

## SpRY DNA cleavage is limited by a reduced rate of R-loop completion

Having established that SpRY indiscriminately binds to DNA, we next analyzed how SpRY interacts with, and cleaves target sequences. To complement the coarse-grained single-molecule imaging approach described in the previous section, we performed kinetic analysis of DNA cleavage by SpRY using ideal short (55-bp) DNA substrates containing a single, perfectly-matched protospacer, under highly controlled in vitro conditions (as has been previously benchmarked for Cas9)[26]. Since SpRY can cleave substrates with all possible PAM sequences[9,16], we focused our efforts on the NGG PAM and two alternative PAM sequences only accessible to SpRY for more detailed kinetic analysis. In this way, we can directly compare SpRY to Cas9, and explore the efficiency of SpRY across diverse sequences.

We first measured the rate of R-loop rate-limited DNA cleavage by mixing a preformed SpRY:gRNA complex with DNA in the presence of $MgCl_2$. We observed that SpRY cleaved the TS at a rate of $0.0025\,s^{-1}$ and the NTS at a rate of $0.0023\,s^{-1}$, ~500-fold slower than the observed rates of Cas9 R-loop rate-limited DNA cleavage[27] (Fig. 4a, c, d). The slow observed rate of cleavage was not unique to the NGG PAM sequence, as SpRY cleaved the TS of NGC and NAC PAM substates with similarly slow rates (Fig. 4a and Supplementary Figs. 5 and 6).

To determine whether the slow rate was due to a reduced rate of the chemical step, or due to a slow step preceding chemistry, we measured cleavage rates after pre-forming the R-loop by preincubating SpRY:gRNA:DNA in the absence of magnesium and then initiating the reaction by adding $MgCl_2$. The reaction was monitored by rapid quench methods since the rates were too fast to resolve by hand mixing. We observed cleavage rates of $6.2\,s^{-1}$ for the TS compared to $4.3\,s^{-1}$ for Cas9[27] (Fig. 4f and Supplementary Figs. 5 and 6). Cleavage of the NTS for SpRY was biphasic with a fast phase at $4.3\,s^{-1}$ with a smaller amplitude reaction at $0.54\,s^{-1}$, compared to an observed rate of $3.5\,s^{-1}$ for Cas9 (Fig. 4g and Supplementary Figs. 5 and 6).

By initiating the reaction with the addition of $Mg^{2+}$ to a preformed enzyme-DNA complex, we bypass slow R-loop formation and measure the intrinsic rate constants for TS and NTS cleavage. We have previously demonstrated that in the presence of $Mg^{2+}$, the rate of HNH

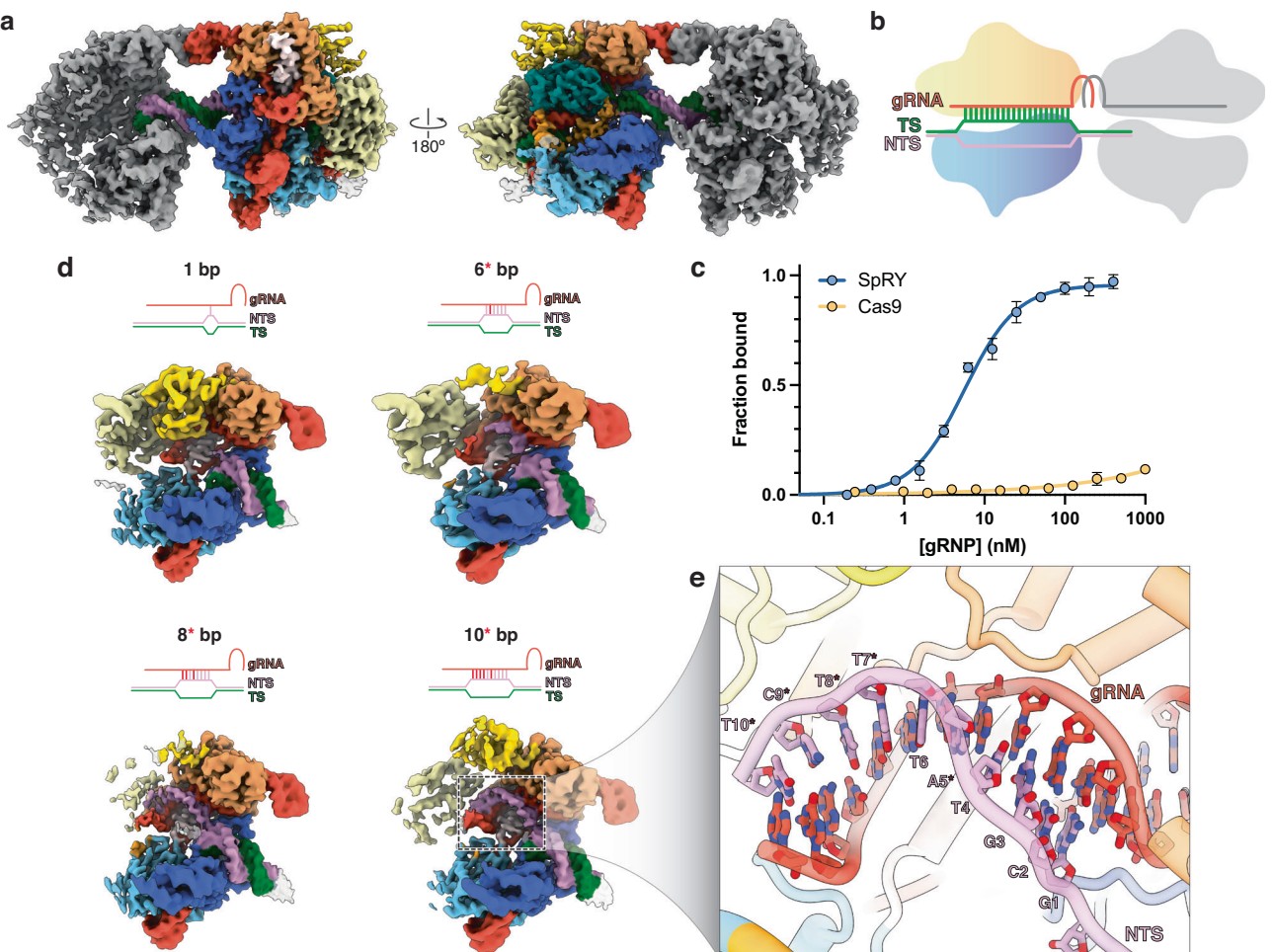

**Fig. 3 | SpRY stably binds to off-target sequences. a** 3.0 Å cryo-EM reconstruction of the SpRYmer with one active SpRY molecule (colored) and one inactive SpRY molecule (gray) bound to the same DNA substrate. **b** Schematic representation of the SpRYmer shown in (**a**). **c** SpRY tightly binds to an untargeted DNA sequence but Cas9 does not. SpRY or Cas9 were titrated into 0.8 nM FAM-labeled off-target DNA substrate. SpRY bound with an apparent $K_d = 6 \pm 0.5$ nM, while for Cas9 $K_d > 8$ μM. Representative of three independent experiments. Error bars indicate standard deviation. **d** Cryo-EM maps of SpRY-gRNA in complex with off-target DNA at different stages of non-productive R-loop formation. Nucleic acid schematics depicted above their corresponding cryo-EM map. *6, 8, and 10 bp structures contain one, three, and five mismatches, respectively. Mismatches are colored red. REC2 and REC3 domains exhibit diffuse density in the 6*, 8* and 10*bp structures. **e** Close-up image of the 10* bp off-target R-loop structure with mismatches indicated by an asterisks. Source data are provided as a Source data file.

repositioning is near-identical to the rate of R-loop completion[26,28]. While we cannot rule out a subsequent conformational change after HNH repositioning upon magnesium addition that limits the observed rate of catalysis, our results reflect a lower limit on the rate of cleavage after R-loop completion. The slower rates observed by monitoring DNA cleavage after mixing enzyme and DNA (in the presence of $Mg^{2+}$) are thought to be limited by the slow rates of R-loop completion.

To test this, we directly measured the rate of DNA unwinding using a stopped-flow assay. A DNA substrate containing the fluorescent tricyclic cytosine (tC°) analog at either position 1 or position 16 on the NTS was used to report on R-loop initiation and completion. As the R-loop propagates, the NTS is displaced resulting in an increase in tC° fluorescence. Paralleling our cleavage measurements, SpRY initiated DNA unwinding (tC° at position 1) of NGG, NGC and NAC substrates with nearly identical observed rates (0.015, 0.022 and 0.046 $s^{-1}$, respectively) (Fig. 4b). R-loop completion (tC° at position 16) was ~3-8-fold slower than initial DNA unwinding (0.004, 0.006, and 0.006 $s^{-1}$, respectively) (Fig. 4b, e). The difference between the rate of unwinding at these two positions support the notion that SpRY unwinds DNA in a directional manner, from PAM-proximal to -distal, akin to Cas9. These data suggest that the decreased observed rate of DNA cleavage by SpRY is due to the decreased rate of R-loop completion.

**Structural basis of target search and R-loop formation by SpRY**

Due to the rapid rate of R-loop formation and DNA cleavage by Cas9, it has only been possible to capture intermediates of this process through covalent crosslinking, DNA substrates containing mismatches, or catalytically dead enzyme[24,29]. We reasoned that the significantly reduced enzyme kinetics may make it feasible to determine time-resolved structures of a catalytically active SpRY in the process of R-loop propagation.

Contrary to typical cryo-EM sample preparation, where samples are optimized for maximal homogeneity and stability, we instead chose to prepare our sample under conditions that enrich for maximal conformational heterogeneity to trap an ensemble of on-pathway intermediate states. Based on our kinetic data, we prepared cryo-EM samples 1 min after mixing SpRY:gRNA with NAC PAM DNA. At this time point, we estimate that ~90% of the DNA is unwound at position 1, much less of the DNA is unwound at position 16, and only ~10% of the DNA is cleaved. This approach captures intermediate states of SpRY as it initiates, propagates, and ultimately completes R-loop formation to achieve DNA cleavage. This cryo-EM dataset yielded seven high-resolution (2.9–3.4 Å) structures of SpRY at distinct states along the directional R-loop propagation pathway: 0, 2, 3, 10, 13, 18 and 20 bp (product state) (Fig. 5a and Supplementary Fig. 7).

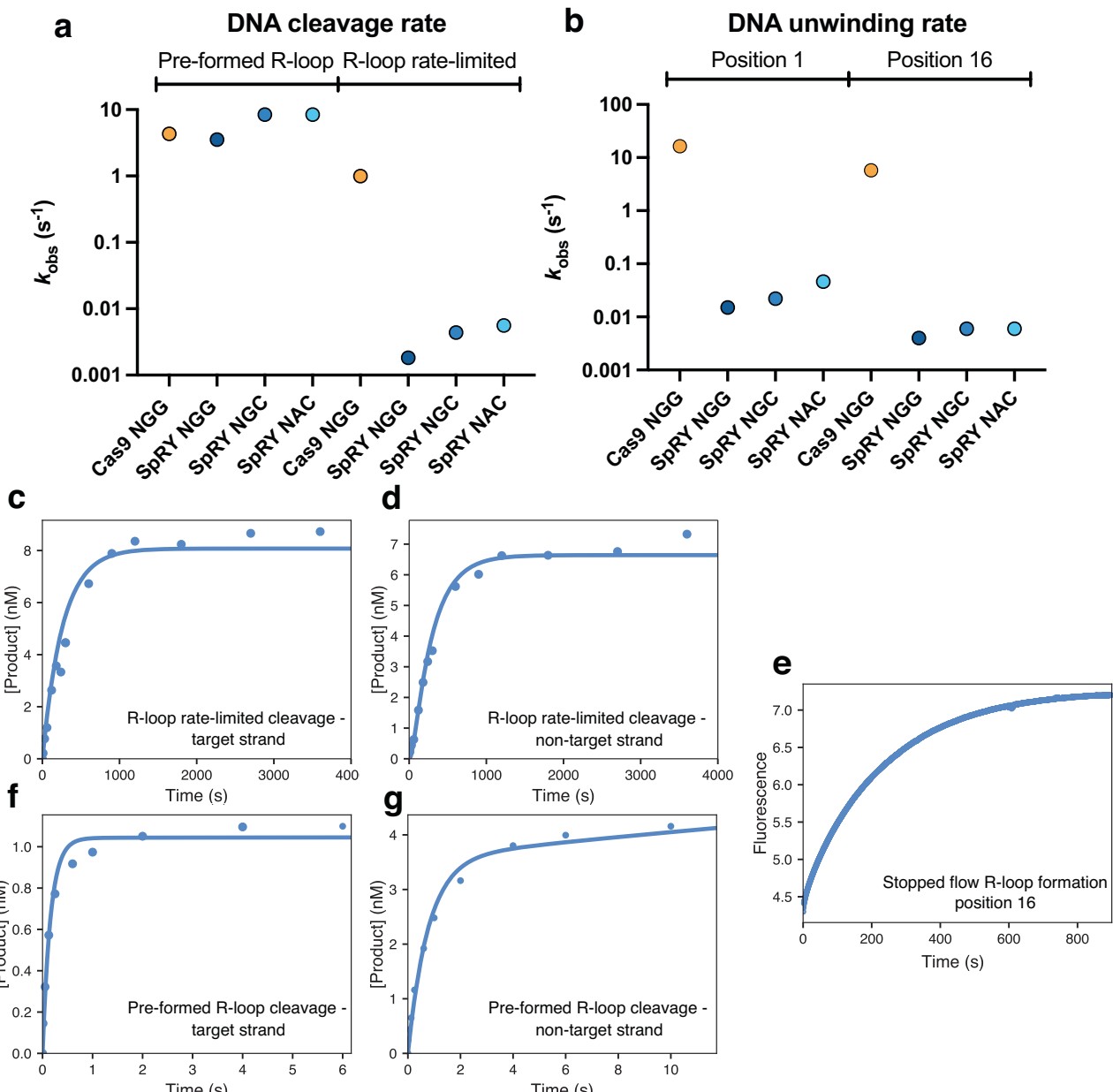

**Fig. 4 | Unfavorable DNA unwinding limits the rate of SpRY target cleavage. a** SpRY and Cas9 observed R-loop rate-limited (DNA-binding initiated) and pre-formed R-loop (Mg²⁺-initiated) cleavage rates. **b** SpRY and Cas9 observed rates of R-loop formation at position 1 (PAM-proximal) and position 16 (PAM-distal). **c** DNA-initiated cleavage of the target strand of DNA. SpRY and sgRNA was mixed with 10 nM DNA in the presence of Mg²⁺ to start the reaction. Fitting the data to a single-exponential function (Supplementary Fig. 5) gives an observed rate of $0.0025 \pm 0.0001\,s^{-1}$. **d** DNA initiated cleavage of the non-target strand of DNA. Reaction conditions were identical to (**c**) but the label was on the non-target strand. Fitting the data to a single-exponential function (Supplementary Fig. 5) gives an observed rate of $0.0023 \pm 0.0001\,s^{-1}$. **e** DNA unwinding was measured by mixing SpRY and sgRNA was with tC° labeled DNA to start the reaction and measured using

stopped flow. Fitting the data to a single-exponential function (Supplementary Fig. 5) gives an observed rate of $0.0044 \pm 0.00001\,s^{-1}$. **f** Mg²⁺ initiated cleavage of the target strand. SpRY, sgRNA, and DNA were mixed with Mg²⁺ to start the reaction using rapid quench. Fitting the data to a single-exponential function (Supplementary Fig. 5) gives an observed rate of $5.4 \pm 0.8\,s^{-1}$. **g** Mg²⁺ initiated cleavage of the non-target strand. Reaction conditions were identical to (**f**) but the label was on the non-target strand. Fitting the data to a double exponential function (Supplementary Fig. 5) gives observed rates of $4.3 \pm 1.3\,s^{-1}$ and $0.54 \pm 0.09\,s^{-1}$ and amplitudes of $1.2 \pm 0.3$ nM and $2.9 \pm 0.3$ nM for the fast and slow phases, respectively. For (**c**–**g**), the solid line through the data points shows the best fit by global data fitting with the model and rate constants summarized in Fig. 7. Source data are provided as a Source Data file.

Since SpRY lacks a PAM requirement to bind DNA targets, it must instead identify targets through probing for sequence complementarity to the gRNA. The 0 bp structure reveals that SpRY interrogates DNA targets by introducing a ~50° kink in the center of the target duplex, distorting the phosphodiester backbone immediately adjacent to the 3' end of the gRNA spacer (Fig. 5a). This structure is highly similar to a recently determined structure of Cas9 covalently crosslinked to non-complementary DNA[24] (Supplementary Fig. 8).

Interestingly, we did not observe an open protein/linear DNA conformation, suggesting that this conformation may be too transient to capture or a product of crosslinking.

The most populated SpRY R-loop intermediate has two bases of the TS base pairing with the gRNA (Fig. 5a). Within our 2 bp structure, we resolved a contact between D269 and position 3 of the gRNA spacer. This contact appears to act as a physical barrier for R-loop propagation, tethering REC2 to the seed region of the gRNA and

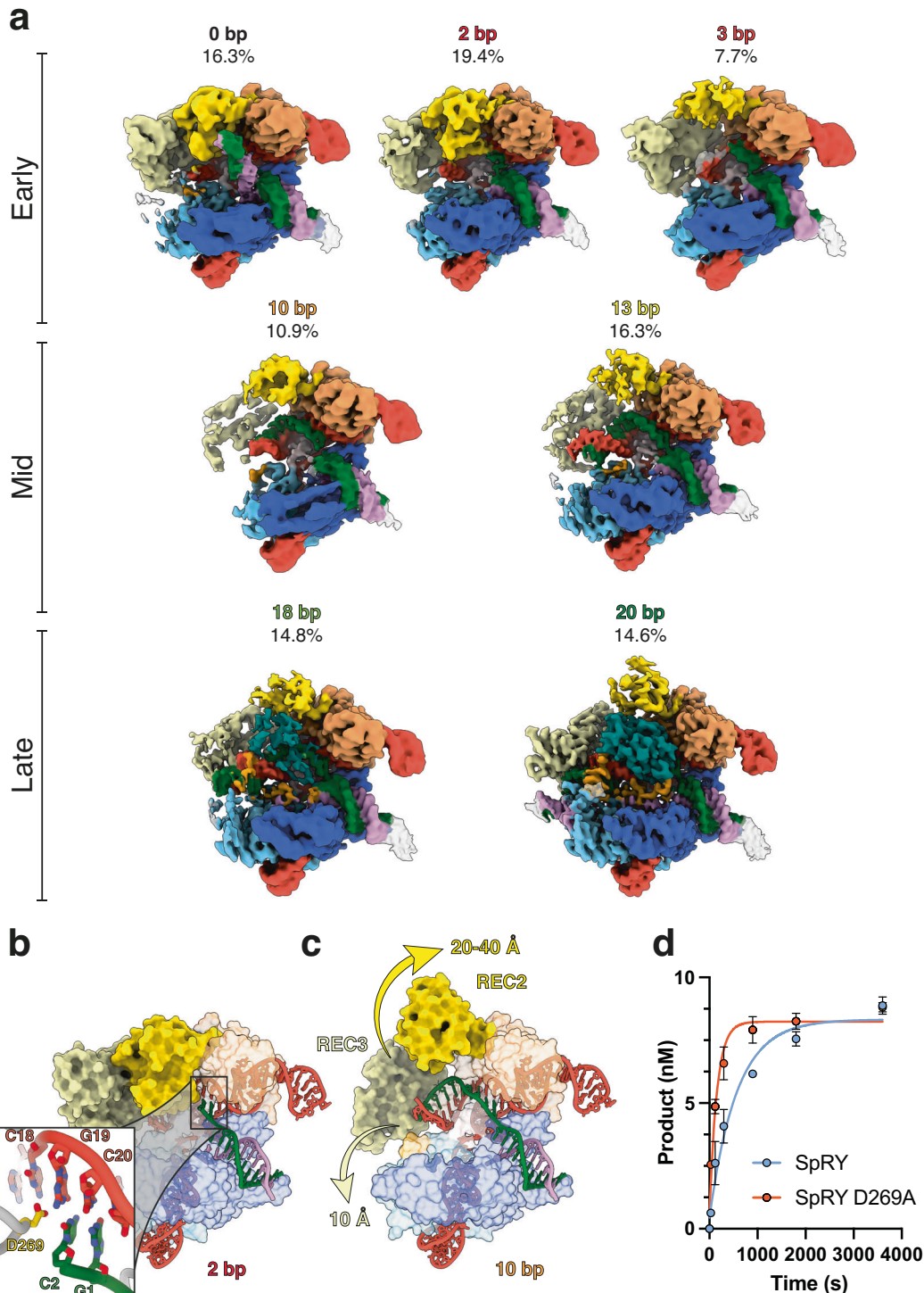

**Fig. 5 | Direct visualization of SpRY R-loop formation. a** Ensemble cryo-EM reconstructions of SpRY during R-loop propagation with the proportion of particles as a percentage. **b, c** Transition from early- to mid-R-loop formation requires a large conformational change of REC2 domain to create a channel to accommodate the R-loop. **d** Alanine substitution of D269 increases the TS cleavage rate from $0.0025\,s^{-1}$ to $0.0065\,s^{-1}$. Representative of three independent experiments. Error bars indicate standard deviation. Source data are provided as a Source Data file.

blocking R-loop propagation (Fig. 5b). Transition to later R-loop intermediates is accompanied by the displacement of REC2 by ~40 Å, creating a channel to accommodate the propagating R-loop (Fig. 5c). Mutation of D269 to an alanine residue increased the rate of TS cleavage by ~4-fold for the NGG PAM, confirming that disruption of this contact accelerates R-loop propagation (Fig. 5d). This physical barrier may be important for probing initial complementarity by making base-

pairing between the gRNA and mismatch-containing duplexes unfavorable. Displacing D269 as the R-loop propagates to 3 bp unlocks REC2 from its binary position and enables further R-loop propagation (Fig. 5a).

The next intermediate we resolved had 10 bp of R-loop formed. In this structure, the REC2 domain has been displaced, shifting upwards by up to 40 Å, while REC3 is poorly resolved due to conformational

flexibility. By 13 bp of R-loop formation, REC2 and REC3 are well-resolved and stabilized by R-loop contacts.

The 18 bp structure has a near-complete R-loop, and exhibits partial density for the HNH domain repositioned at the scissile phosphate of the TS. When we examined the catalytic sites, we found that cleavage had not occurred at either HNH or RuvC. There is density for the required $Mg^{2+}$ ion in the HNH active site, yet at the RuvC active site, there is only density for one of the two required $Mg^{2+}$ ions. This structure represents a state just prior to cleavage where one of the $Mg^{2+}$ ions is not yet properly coordinated for cleavage.

The distribution of R-loop duplex lengths we captured represent kinetic barriers for R-loop propagation. Large conformational changes are associated with each observed intermediate. While the 0 bp and 2 bp structures largely resemble the Cas9 binary complex (RMSD 1.88 Å), the transition to the 3 bp structure requires the seismic eviction of the REC2 domain. Propagation to 10 bp necessitates movement of the REC3 domain, evidenced by the lack of REC3 density in our reconstruction. By 18 bp of heteroduplex formation, HNH begins docking into the active state leading to DNA cleavage as the R-loop completes 20 bp. Together these structures support a model of R-loop formation distinguished by structural intermediates, as previously proposed and now observed under active enzyme conditions[15,17,24,29]. Furthermore, the 2 bp and 13 bp intermediates provide additional evidence for this model, yet had not been previously observed.

### SpG is a functional intermediate between Cas9 and SpRY

Having established the molecular framework of DNA targeting by SpRY, we then aimed to elucidate the mechanisms governing the behavior of its precursor, SpG-Cas9, which has an NGN PAM preference[16]. Since SpRY and SpG share six mutations (D1335L, S1136W, G1218K, E1219Q, R1335Q, and T1337R) (Fig. 6a), we hypothesized that SpG may be a potential intermediate between Cas9 and SpRY in terms of PAM promiscuity and DNA cleavage efficiency. While SpRY loses all PAM specificity, SpG retains specificity for guanine at position 2 of the PAM through residue R1333. To probe whether SpG and SpRY operate through analogous mechanisms of engaging diverse PAM sequences, we determined the structures of SpG bound to NGG and NGC PAM-containing DNA substrates with global resolutions of 3.1 Å and 3.6 Å, respectively (Fig. 6b, c and Supplementary Fig. 9).

Through structural analysis of SpG, we uncovered that four mutations (D1135L, S1136W, E1219Q, and T1337Q) engender DNA duplex interactions mirroring those delineated for SpRY (Fig. 6b–d). Intriguingly, the mutations G1218K and R1335Q, which adopt different rotamers for different PAM sequences in SpRY, maintain the same rotamer conformation in the context of SpG. This observation raises the possibility that R1333 in SpG is pre-ordered and poised for rapid PAM recognition similar to Cas9. Therefore, SpG recognizes DNA targets through a hybrid mechanism of base-specific and non-specific electrostatic contacts that resemble both Cas9 and SpRY (Fig. 6e). This is also similar to the mechanism of SpCas9-NG[30].

We then investigated the DNA cleavage kinetics of SpG across all NGN PAM sequences. R-loop rate-limited DNA cleavage by SpG occurs at rates of 0.04 $s^{-1}$ and 0.03 $s^{-1}$ for the TS and NTS, respectively (Supplementary Fig. 10). These cleavage rates are ~20-fold faster than those observed for SpRY, underscoring the tradeoff between expanded targeting capacity and cleavage efficiency. Conversely, the pre-formed R-loop cleavage rates for SpG align with those of Cas9 with observed rates of 12.8 $s^{-1}$ and 1.1 $s^{-1}$ for TS and NTS cleavage, respectively (Supplementary Fig. 10). Together, these observations strongly suggest that akin to SpRY, SpG also grapples with a deficiency in R-loop formation, albeit to a lesser extent.

We then measured the rates of DNA unwinding for both NGG and NGC PAM substrates using the same stopped-flow assay described above for SpRY. SpG initiated DNA unwinding of NGG and NGC substrates with observed rates of 0.44 $s^{-1}$ and 0.47 $s^{-1}$ respectively

(Supplementary Fig. 10). Consistent with Cas9 and SpRY, R-loop completion for SpG was slower than initiation with observed rates of 0.056 $s^{-1}$ and 0.063 $s^{-1}$ for the NGG and NGC substrates, respectively (Supplementary Fig. 10). Hence, the reduced DNA cleavage rates by SpG in comparison to Cas9, stem from stifled DNA unwinding rates. However, the impairment in R-loop completion is more pronounced in SpRY than in SpG. Our comprehensive analyses illuminate distinct facets of SpG's mode of action, positioning it as an intermediate between the benchmarks set by Cas9 and SpRY.

### Global fitting reveals unfavorable R-loop completion is a major difference between SpRY, SpG, and Cas9

Observed rates are often complex functions of multiple intrinsic rate constants which can be difficult to accurately interpret on their own[31–33]. When a mutant enzyme is generated, the rate constants of the wild-type kinetic pathway are altered. Therefore, we fit our kinetic measurements by simulation to a model that accounts for all the data to decipher how the rate constants diverge between SpRY, SpG, and Cas9.

To fully understand the mechanisms of SpRY and SpG in comparison to Cas9, we globally fit all the experiments of SpRY (Fig. 4c–g) and SpG (Fig. 6e–i) binding and cleaving the NGG PAM DNA based on the model previously derived for Cas9 (Supplementary Fig. 12). A comparison of the rate constants and free energy profiles for SpRY, SpG, and Cas9 are given in Fig. 7 (and supported by confidence contour analysis (Supplementary Fig. 11 and Supplementary Table 3).

Although the forward ($k_2$) and reverse ($k_{-2}$) rate constants, and equilibrium constant ($K_2$) for R-loop formation were individually well resolved for Cas9 ($k_2 = 2.5$ $s^{-1}$, $k_{-2} = 1.2$ $s^{-1}$, $K_2 = 2$)[20], for both SpRY and SpG, we found that R-loop formation is unfavorable and rapidly reversible. Global fitting of the data for SpRY places lower limits on R-loop formation and R-loop collapse of $k_2 > 0.02$ $s^{-1}$ and $k_{-2} > 20$ $s^{-1}$, respectively, with an equilibrium constant $K_2 = 0.0001$. For SpG, global fitting of the data places an equilibrium constant of R-loop formation $K_2 = 0.0005$ with rates of R-loop formation and decay between Cas9 and SpRY, at $k_2 = 0.3$ and $k_{-2} = 60$ $s^{-1}$. The observed rate of R-loop formation for SpRY (0.00437 $s^{-1}$) measured by stopped flow can be reconciled with the intrinsic rate of cleavage (5.4 $s^{-1}$) using global fitting of the data. The model implies that the observed cleavage rate, $k_{obs} = K_2 k_3 = 0.005$ $s^{-1}$, where $k_3$ is the intrinsic rate constant for HNH cleavage and $K_2$ is the fraction of enzyme poised for catalysis. Thus, neither of the observed rates for R-loop formation and DNA cleavage by SpRY (Fig. 2c, e) actually measure the individual steps. Rather, for both SpRY and SpG the equilibrium for R-loop formation is unfavorable and is pulled forward by subsequent the DNA cleavage step. Since the favorable equilibrium for R-loop formation by Cas9 increases its efficiency but limits its specificity by slow dissociation from off-target sequences, readily reversible R-loop formation could lead to increased fidelity by allowing off-target DNA to be released.

The free energy profile shows that the major difference between SpRY, SpG, and Cas9 is the largely unfavorable R-loop completion step, which reduces the observed rates of cleavage (Fig. 7a). Notably, R-loop completion for SpG is more favorable than it is for SpRY, but less favorable than it is for Cas9, congruent with our kinetic findings that place SpG as a functional intermediate. This rigorous kinetic analysis shows that R-loop completion is thermodynamically unfavorable, yet the equilibrium is pulled forward by, and kinetically linked to the subsequent irreversible HNH cleavage step (Fig. 7b). This unfavorable equilibrium allowed us to capture multiple intermediates of SpRY using kinetics-guided structural studies, and elucidate the underlying mechanism of PAMless target search (Fig. 7c).

## Discussion

While CRISPR-Cas9 has found widespread use as a powerful tool for programmable gene editing, the identification of target sequences by

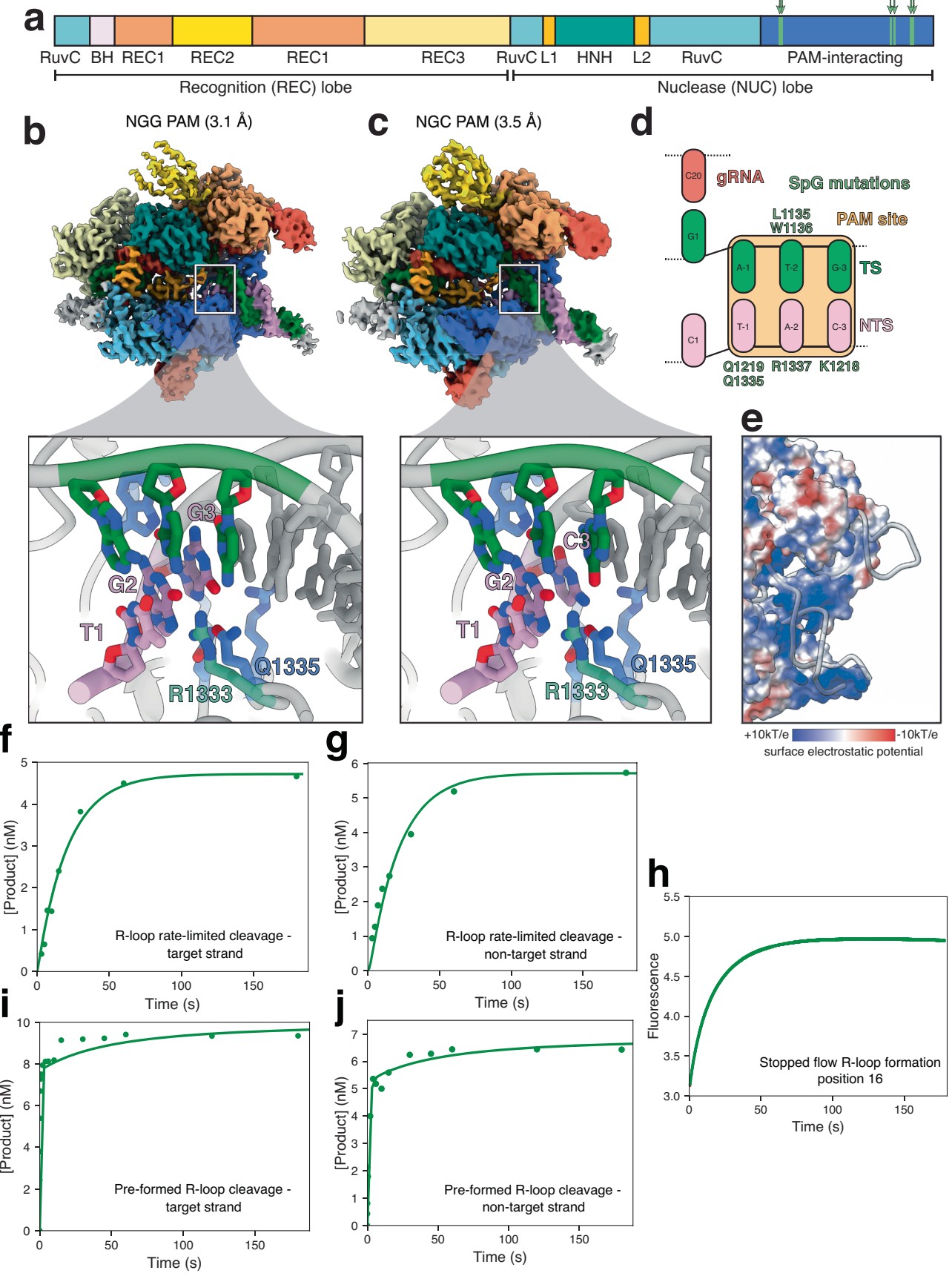

**Fig. 6 | SpG targets NGN PAM sequences through base-specific and electrostatic DNA contacts. a** Cas9 domain organization with SpG mutations highlighted in green. **b** 3.1 Å cryo-EM reconstruction of SpG bound to target DNA with NGG PAM sequence. **c** 3.5 Å cryo-EM reconstruction of SpG bound to target DNA with NGC PAM sequence. Detailed PAM site interactions are shown below for each

reconstruction. **d** Schematic representation of the SpG PAM site interaction. **e** Surface electrostatic potential map of SpG. **f–j** Global fit of kinetic data for SpG. Experiments were performed as described for SpRY in Fig. 2. Rate constants from the resulting fit are shown in Fig. 7. Source data are provided as a Source Data file.

Cas9 presents a formidable challenge. The search predicament is simplified by the NGG PAM requirement, which significantly reduces the number of sequences that are probed for gRNA complementarity. Without PAM specificity, SpRY must initiate DNA melting at far more sequences to locate gRNA complementarity. Considering the topological complexity of chromatin[34–36], SpRY is likely to exhibit a preference for targeting regions with low energetic barriers of DNA melting such as bends and active genes as was previously suggested[37,38]. Our findings shed light on the mechanism of SpRY targeting, and its propensity for specific genomic locations.

The recycling of Cas9 in vivo is facilitated by the recruitment of chromatin remodelers and DNA-repair factors after cleavage[3,39–41]. Unlike Cas9, SpRY stably associates with off-target sequences, as revealed by ensemble biochemistry and single-molecule imaging experiments. Congruent with our findings, live-cell, single-molecule tracking recently demonstrated that SpRY exhibits increased binding at off-target sites and takes nearly double the amount of time to find the correct target site compared to Cas9[23]. The accumulation of SpRY at off-target sites results in a reduction in the effective concentration of active enzyme until it is displaced by cellular factors[42]. Although this phenomenon may not drastically increase the occurrence of off-target double-strand breaks due to insufficient gRNA complementarity necessary for HNH realignment and DNA cleavage[17], it does have the potential to induce undesirable gene silencing[37,43]. Hence the importance of carefully considering the implications of SpRY and Cas9 behavior to mitigate off-target effects and enhance the safety of gene editing applications.

The ultimate gene editor would exhibit expanded targeting capacity, as SpRY does, with high fidelity and high efficiency. Our multimodal approach toward dissecting the targeting mechanisms of SpRY has proven that broadening target sequence space comes at the cost of reduced efficiency. The wealth of Cas9 research bestows countless mutations to tweak Cas9 for any application. SpRY in combination with known hyperactive mutations may increase the efficiency of SpRY without compromising fidelity[44]. Striking a balance between expanded sequence space, fidelity, and efficiency is pertinent to consider when applying or engineering gene editing technologies.

In this way, SpG emerges as a promising intermediate between Cas9 and SpRY by retaining PAM specificity for a single guanine while exhibiting reasonable DNA cleavage efficiency. Our structural analysis of SpG revealed that it engages diverse NGN PAM sequences through a hybrid mechanism combining base-specific and electrostatic contacts. The mode of action by SpG solidifies its role as a valuable tool for precision gene editing.

Despite its limitations in cleavage efficiency, SpRY proves to be an asset in modifying genomic regions that were previously inaccessible to Cas9. Through single-molecule imaging, cryo-EM, and enzyme kinetics we provide mechanistic insight into how SpRY targets diverse sequences. Using active enzyme with bona fide substrates, our structures represent on-pathway intermediates of SpRY. Furthermore, our structures were resolved from kinetics-guided ensemble reactions which capture states likely corresponding to physiologically-relevant bottlenecks during SpRY activation. Surmounting the constraints presented here through the creation of chimeric CRISPR-Cas9 variants could forge the development of highly specific and adaptable gene editing tools that transcend biological constraints. Our discoveries will serve as a valuable foundation for the generation of potent chimeric Cas9 variants advancing the field toward the most effective gene editing tool.

## Methods

### Protein expression and purification
Cas9 was expressed and purified as described previously[26]. SpRY was expressed from pET28-SpRY-NLS-6xHis (gbr2101) purchased from Addgene (Addgene plasmid # 181743; http://n2t.net/addgene:181743; RRID:Addgene_181743)[9]. SpRY was then expressed and purified in the same manner as Cas9. SpG mutations were introduced via PCR mutagenesis to the Cas9 plasmid, and enzyme was purified as described for Cas9 and SpRY.

### Nucleic acid preparation
Fifty-five-nt DNA duplexes were prepared from PAGE-purified oligonucleotides synthesized by Integrated DNA Technologies. DNA duplexes used in cleavage assays were prepared by mixing 6-FAM or Cy3 labeled target strands with unlabeled non-target strands at a 1:1.15 molar ratio in annealing buffer (10 mM Tris-HCl pH 8, 50 mM NaCl, 1 mM EDTA), heating to 95 °C for 5 min, then cooling to room temperature over the course of 1 h. The sgRNA was purchased from Synthego and annealed in annealing buffer using the same protocol as for the duplex DNA substrates. The sequences of the synthesized oligonucleotide are listed in Supplementary Table 1.

### Buffer composition for kinetic reactions
Cleavage reactions were performed in 1X cleavage buffer (20 mM Tris-Cl, pH 7.5, 100 mM KCl, 5% glycerol, 1 mM DTT) at 37 °C.

### DNA cleavage kinetics
Pre-formed R-loop DNA cleavage: The reaction of Cas9 with on- and off-target DNA was performed by preincubating Cas9.gRNA (28 nM active-site concentration of Cas9, 100 nM gRNA) with 10 nM DNA with a 6-FAM label on the target strand in the absence of $Mg^{2+}$, and trace amounts of EDTA to chelate any co-purified cations (0.2 mM). The reaction was initiated by adding $Mg^{2+}$ to 10 mM, then stopped at various times by mixing with 0.3 M EDTA (Supplementary Fig. 1). R-loop rate-limited DNA cleavage: as above, but Cas9, gRNA and $Mg^{2+}$ were co-incubated, and the reaction was initiated by the addition of 10 nM DNA. Products of the reaction were resolved and quantified using an Applied Biosystems DNA sequencer (ABI 3130xl) equipped with a 36 cm capillary array and nanoPOP6 polymer (MCLab)[45]. Data fit to equations were fit using either a single or double-exponential equations shown below:

Single exponential equation:

$$Y = A_1 e^{-\lambda_1 t} + C \qquad (1)$$

where $Y$ represents concentration of cleavage product, $A_1$ represents the amplitude, $\lambda_1$ represents the observed decay rate (eigenvalue) and $C$ is the endpoint. The half-life was calculated as $t_{1/2} = \ln(2)/\lambda_1$.

Double exponential equation:

$$Y = A_1 e^{-\lambda_1 t} A_2 e^{-\lambda_2 t} + C \qquad (2)$$

where $Y$ represents concentration of cleavage product, $A_1$ represents the amplitude and $\lambda_1$ represents the observed rate for the first phase. $A_2$ represents the amplitude and $\lambda_2$ represents the observed rate for the second phase, and $C$ is the endpoint.

### Stopped-flow kinetic assay
Stopped-flow experiment was performed as previously described[26]. Briefly, 250 nM Cas9-gRNA complex (1:1 ratio, active site Cas9

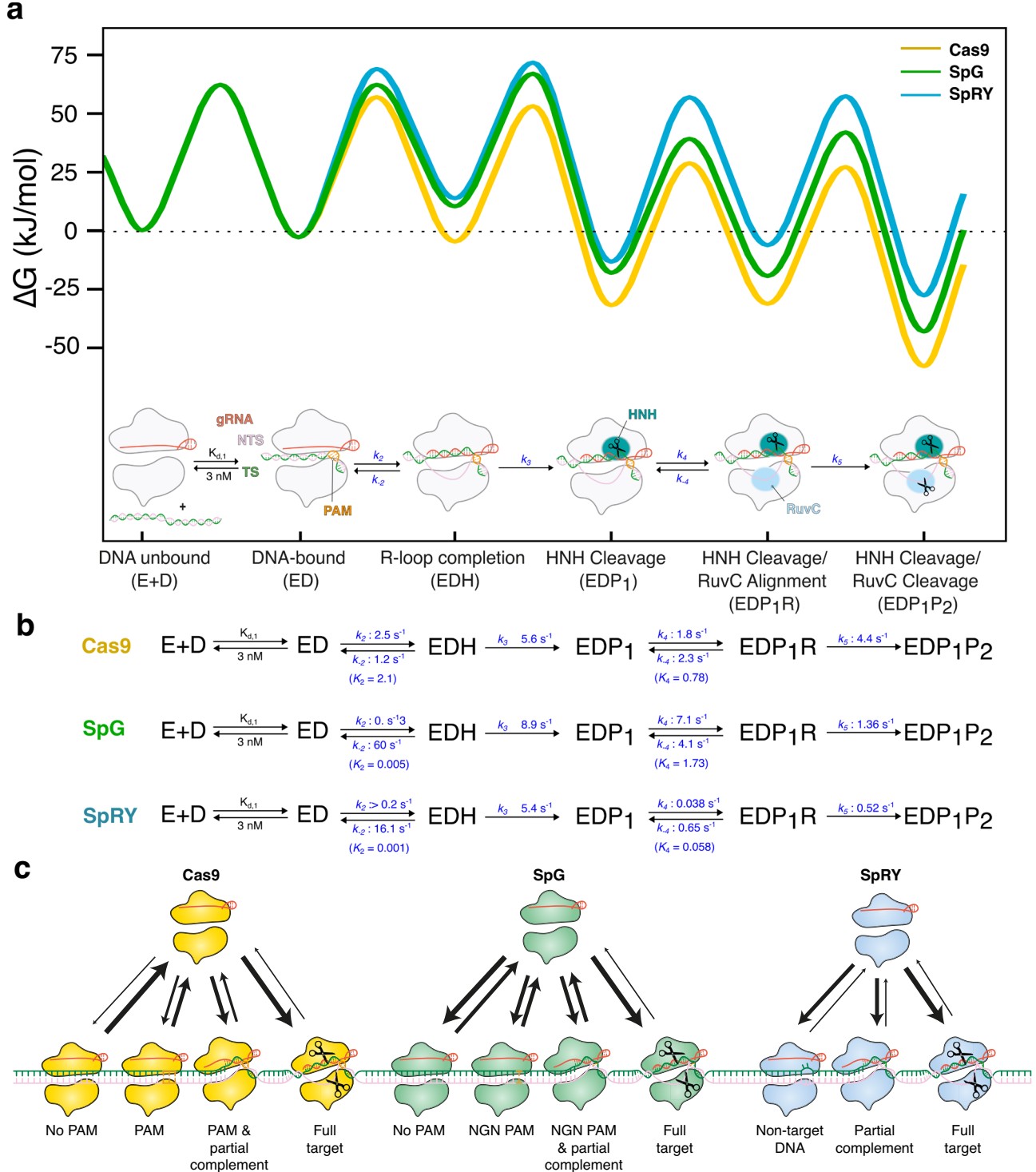

**Fig. 7 | The kinetic barrier toward R-loop completion distinguishes Cas9, SpG, and SpRY. a** Free energy profile for SpRY, SpG, and Cas9 enzyme cleavage. Profiles are shown in blue for SpRY, green for SpG, and yellow for Cas9. **b** Detailed kinetic scheme along with a cartoon of the enzyme reaction pathway derived from global fitting calculated as described[38–40]. In prior work on Cas9 we used a branched pathway to allow RuvC NTS cleavage before or after HNH TS cleavage, for clarity here we only show the kinetically preferred pathway with HNH before RuvC although the data were fit using the branched pathway (Supplementary Fig. 12). **c** Schematic model of target search by Cas9, SpG, and SpRY. Source data are provided as a Source Data file.

concentration) was mixed with 100 nM tC°-labeled 55/55 nt DNA substrate at 37 °C using AutoSF-120 stopped-flow instrument (KinTek Corporation, Austin, TX). Excitation was at 367 nm, and emission was monitored with a 445 nm filter with a 20 nm bandpass (Semrock).

**Global fitting of kinetic data**

Global data fitting was performed by fitting all experiments in KinTek Explorer[46,47] to the reaction scheme in Supplementary Fig. 12, with experimental details of mixing steps and reactant concentrations input for each experiment. In fitting data by simulation, each experiment is

modeled exactly as it was performed. Chemical-quench experiments were modeled simply as the sum of species containing product that was labeled in the experiment (i.e., for RuvC: EDP1 + EDP1R + EDP1P2). Fluorescence transients were modeled by using fluorescence scaling factors.

For example, the experiment shown in Fig. 2e was modeled as:

$$a*(D + ED + (b*(EDH + EDHR + EDHP2 + EDP1 + EDP1R + EDP1P2))) \quad (3)$$

where $a$ scales the overall signal relative to enzyme concentration and $b$ represents the fractional change in fluorescence in forming the R-loop completed state. In the process of fitting data, a value of $b = 1.62$ was derived for SpRY indicating a 62% increase in fluorescence. For SpG, the fitting gave a value of $b = 1.56$.

For second-order DNA binding step ($k_1$), the rate was not defined by the data so the binding rate constant was locked at 1000 $\mu M^{-1} s^{-1}$ and the reverse rate constant was locked at 0.3 $s^{-1}$ to give a $K_d$ for DNA binding of 3 nM, similar to other estimates for the equilibrium constant for Cas9 binding to DNA and rates that are much faster than subsequent R loop formation rates.

The confidence contours were derived using the FitSpace[48] function. These confidence contour plots are calculated by systematically varying a single rate constant and holding it fixed at a particular value while refitting the data allowing all other rate constants to float. The goodness of fit was scored by the resulting $\chi^2$ value. The confidence interval is defined based on a threshold in $\chi^2$ calculated from the F-distribution based on the number of data points and number of variable parameters to give the 95% confidence limits. For the data (Fig. 2), this threshold was of 0.99 and was used to estimate the upper and lower limits for each rate constant.

The free-energy profile in Fig. 2 was created in KinTek Explorer using the rate constants given in Supplementary Table 3 using simple transition state theory.

$$\text{rate} = A * \frac{k_B T}{H} \exp\left(-\triangle G^{\ddagger}/RT\right) \quad (4)$$
$$0 < A \leq 1$$

where $k_B$ is the Boltzmann constant, $h$ is the Planck's constant, and $R$ is the gas constant. The free-energy profile was created using a transmission coefficient of $A = 0.01$ to better show the relationships. A value for DNA concentration of 10 nM was used in the calculation. SpG kinetics and global fitting were performed as described for SpRY.

**Cryo-EM sample preparation, data collection, and processing**

SpRY bound to different DNA substrates were assembled by mixing SpRY with gRNA in a 1:1.5 molar ratio and incubated at room temperature for 10 min in reaction buffer (20 mM Tris-Cl, pH 7.5, 100 mM KCl, 10 mM MgCl$_2$, 5% glycerol, and 5 mM DTT). Each DNA substrate was then added in a 1:1 molar ratio with 10 $\mu M$ SpRY gRNP. The product state PAM complexes were incubated for 1 h at room temperature and the off-target DNA complex was incubated for 2 h at room temperature. Based on our kinetic analysis, R-loop intermediates were prepared by triggering the DNA cleavage reaction using NAC PAM DNA and incubating with SpRY gRNP at 37 °C for 60 s. The reactions were quenched by vitrification. 2.5 $\mu$l of sample was applied to glow discharged holey carbon grids (Quantifoil 1.2/1.3), blotted for 6 s with a blot force of 0, and rapidly plunged into liquid nitrogen-cooled ethane using an FEI Vitrobot MarkIV.

All datasets apart from NTC PAM were collected on an FEI Titan Krios cryo-electron microscope equipped with a K3 Summit direct electron detector (Gatan, Pleasanton, CA). Images were recorded with SerialEM v4.1[49] with a pixel size of 0.83 Å. Movies were recorded at 13.3 electrons/pixel/second for 6 s (80 frames) to give a total dose of 80 electrons/pixel. The NTC dataset was collected on a FEI Glacios cryo-TEM equipped with a Falcon 4 detector with a pixel size of 0.94 Å, and a total exposure time of 15 s resulting in a total accumulated dose of

40 e/Å$^2$ which was split into 60 EER fractions. All datasets were collected with a defocus range of −1.5 to −2.5 $\mu$m. Motion correction, CTF estimation and particle picking was performed on-the-fly using cryoSPARC Live v4.0.0-privatebeta.2[50]. Further data processing was performed with cryoSPARC v.3.2. A total of 2016 movies were collected for the NTC dataset, 4293 movies for the NGG dataset, 3546 movies for the NAC dataset, 10,151 movies for the R-loop intermediate dataset, and 8765 movies for the off-target DNA dataset.

The NTC, NGG, and NAC datasets were processed using similar workflows starting with blob picker with a minimum particle diameter set to 100 Å, and a maximum particle diameter set to 180 Å. The particles were then subjected to a single round of 2D classification. The particles selected from 2D classification were processed via ab initio reconstruction, followed by heterogeneous refinement. After multiple rounds of ab initio reconstruction and heterogeneous refinement, the final reconstructions were formed from non-uniform refinement. There appeared to be a second SpRY molecule bound to the same DNA in some of the NAC PAM 2D classes. Particles within these classes were extracted with a larger box size of 512 pixels yielding a dimer structure. Masks for each half of the dimer were created and particle subtraction was performed on each half. The subtracted particles were then subjected to local refinement, and ultimately recombined in ChimeraX producing the SpRYmer structure.

The R-loop intermediate and off-target DNA datasets were processed following similar workflows, again starting with blob picker as in the previous datasets. After the initial round of 2D classification, ab initio reconstruction, and heterogeneous refinement, the particles were processed using 3D classification with the parameters of force hard classification and PCA initialization mode. This method generated classes with a variable number of R-loop base pairs formed. The particles corresponding to a single state were extracted with a 384-pixel box size, globally and locally CTF refined, and fed to non-uniform refinement which yielded the final reconstructions.

For SpG, DNA-bound gRNP complexes were assembled as described above for SpRY, and cryo-EM samples were prepared after 1 h incubation at room temperature. Samples were applied to grids and vitrified as described above. Cryo-EM data was collected on a Titan Krios microscope as described above, apart from the stage tilted to −30° to compensate for preferred orientation of SpG complexes. NGG- and NGC PAM SpG complex datasets were processed as described for SpRY, with the addition of motion correction performed using MotionCor2 v1.6.4[45]. Dose-weighted, motion-corrected micrographs were imported into cryoSPARC and subsequent data processing was performed as described above.

**Structural model building and refinement**

Product state Cas9 (PDB 7S4X) was used as a starting model for the NGG PAM structure, and the SpG structures. Once built, the NGG PAM structure was used as a starting model for the NAC, NTC, 10 bp, and 18 bp structures. The 18 bp structure was used as a starting model for the 13 bp structure. SpCas9 with 0 bp, closed-protein/bent-DNA conformation (PDB 7S36) was used as a starting model for the 0 bp intermediate. SpCas9 with 3 bp R-loop (PDB 7S38) was used as a starting model for the 2 bp structure, which was then used as the starting model for the 3 bp R-loop intermediate and 1 bp off-target structure. The 3 bp R-loop intermediate was used as a starting model for the 6* bp off-target structure, which was then used as the starting model for the 8* bp off-target structure. The 10 bp R-loop intermediate was used as the starting model for the 10* bp off-target structure. SpCas9 binary complex (PDB 4ZT0) was used as the starting model for the inactive half of the SpRYmer, and the NAC PAM structure created for this manuscript was used as the starting model for the active half. Nucleic acid alterations were made in Coot v1.1.07, and further modeling was performed using Isolde v1.6[51]. The models were ultimately subjected to real-space refinement implemented in Phenix v1.21.

All structural figures and videos were generated using ChimeraX v1.2[52].

## Single-molecule fluorescence microscopy

Single-molecule fluorescent images were collected using a customized prism TIRF microscopy-based inverted Nikon Ti-E microscope system equipped with a motorized stage (Prio ProScan II H117). The sample was illuminated with a 488 nm laser (Coherent Sapphire) through a quartz prism (Tower Optical Co.). For imaging SYTOX Orange-stained DNA and Anti-FLAG-Qdot705-labled dSpRY or dSpCas9, the 488 nm laser power was adjusted to deliver low power (4 mW) at the front face of the prism using a neutral density filter set (Thorlabs). The imaging was recorded using electron-multiplying charge-coupled device (EMCCD) cameras (Andor iXon DU897). Flowcells used for single-molecule DNA experiments were prepared as previously described[53]. Briefly, a 4-mm-wide, 100-μm-high flow channel was constructed between a glass coverslip (VWR 48393 059) and a custom-made quartz microscope slide using two-sided tape (3 M 665). Double-tethered DNA curtains were prepared with 40 μl of liposome stock solution (97.7% DOPC (Avanti #850375P), 2.0% DOPE- mPEG2k (Avanti #880130P), and 0.3% DOPE-biotin (Avanti #870273P) in 960 μl Lipids Buffer (10 mM Tris-HCl, pH 8, 100 mM NaCl) incubated in the flowcell for 30 min. Then, 50 μg μl$^{-1}$ of anti-rabbit polyclonal antibody (ICL Labs, #GGHL-15A) diluted in Lipids Buffer was incubated in the flowcell for 10 min. The flowcell was washed with BSA Buffer (40 mM Tris–HCl, pH 8, 2 mM MgCl2, 1 mM DTT, 0.2 mg ml$^{-1}$ BSA) and 1 μg l$^{-1}$ of digoxigenin monoclonal antibody (Life Technologies, #700772) diluted in BSA Buffer was injected and incubated for 10 min. Streptavidin (0.1 mg ml$^{-1}$ diluted in BSA Buffer) was injected into the flowcell for another 10 min. Finally, -12.5 ng μl$^{-1}$ of DNA substrate was injected into the flowcell. To prepare single-tethered DNA curtains, the anti-rabbit antibody and digoxigenin antibody steps were omitted.

For single-tethered DNA experiments, DNA was visualized in the presence of a continuous flow (0.2 ml min$^{-1}$) in the imaging buffer (40 mM Tris-HCl pH 8.0, 2 mM MgCl2, 0.2 mg ml$^{-1}$ BSA, 50 mM NaCl, 1 mM DTT, 100 nM SYTOX Orange) supplemented with an oxygen scavenging system (3% D-glucose (w/v), 1 mM Trolox, 1500 units catalase, 250 units glucose oxidase). For double-tethered DNA experiments, the flow was turned off after protein entered into flowcell. NIS-Elements software (Nikon) was used to collect the images at 500 ms frame rate with 100 ms exposure time. All images were exported as uncompressed TIFF stacks for further analysis in FIJI v2.15.1 (NIH) and MATLAB r2024a (The MathWorks). The half lifetime of dSpRY and dSpCas9 were fitted by single-exponential equation.

## DNA substate preparation for single-molecule assays

To prepare DNA substates for single-molecule imaging, 125 μg of λ-phage DNA was mixed with two oligos (2 μM oligo Lab07 and 2 μM oligo Lab09) in 1× T4 DNA ligase reaction buffer (NEB B0202S) and heated to 70 °C for 15 min followed by gradual cooling to 15 °C for 2 h. One oligo will be annealed with the overhand located at the left cohesive end of DNA, and the other oligo will be annealed with the overhand at right cohesive end. After the oligomer hybridization, 2 μl of T4 DNA ligase (NEB M0202S) was added to the mixture and incubated overnight at room temperature to seal nicks on DNA. The ligase was inactivated with 2 M NaCl, and the reaction was injected to an S-1000 gel filtration column (GE) to remove excess oligonucleotides and proteins.

## dSpRY-gRNA and dSpCas9-gRNA preparation and fluorescent labeling

dSpRY and dSpCas9 ribonucleoprotein complexes were reconstituted by incubating a 1:2 molar ratio of apoprotein and sgRNA (sequence: GUG AUA AGU GGA AUG CCA UGG UUU UAG AGC UAG AAA UAG CAA GUU AAA AUA AGG CUA GUC CGU UAU CAA CUU GAA AAA GUG GCA

CCG AGU CGG UGC UUU U, λ target-18.1 kb) in incubation buffer (50 mM Tris-HCl pH 8.0, 5 mM MgCl$_2$, 2 mM DTT) followed by incubation at 25 °C for 25 min. dSpRY and dSpCas9 were labeled by anti-FLAG antibody coupled with Qdot705, and then diluted to 0.5 nM–2 nM in imaging buffer before injected into the flowcell. For dSpRY-gRNA and dSpCas9-gRNA binding experiments, single-tethered DNA curtains were assembled and protein was diluted and incubated with sgRNA as described above. Ribonucleoprotein was diluted in imaging buffer to a final concentration of 2 nM. For dSpRY-gRNA and dSpCas9-gRNA binding lifetime experiments, double-tethered DNA curtains were assembled and ribonucleoprotein was diluted in imaging buffer to a final concentration of 0.5 nM.

## Fluorescence anisotropy binding assays

SpRY and Cas9 gRNP complexes were prepared as described above, and serial 2-fold dilutions were incubated with 0.8 nM FAM-labeled scrambled DNA for 3 h at 37 °C in 1 x Cas9 buffer supplemented with 0.05% Tween-20. Data were recorded at 37 °C in a CLARIOstar Plus multi-detection plate reader (BMG Labtech) equipped with a fluorescence polarization optical module ($\lambda_{ex}$ = 485 nm; $\lambda_{em}$ = 520 nm). The data were fit using a one-step binding model in KinTek Explorer to define the $K_d$ and the extrapolated starting ($Y_0$) and ending ($\Delta Y$) points, which were used to normalize the data for calculation of fraction bound (Fig. 3c). The value of $\Delta Y$ from the SpRY data was also used to normalize the Cas9 data:

$$Y = Y_0 + \Delta Y \cdot \theta \tag{5}$$

and

$$\text{fraction bound } \theta = (Y - Y_0)/\Delta Y = [\text{gRNP}]/(K_d + [\text{gRNP}]) \tag{6}$$

## Reporting summary

Further information on research design is available in the Nature Portfolio Reporting Summary linked to this article.

## Data availability

The structures and associated atomic coordinates have been deposited into the Electron Microscopy Data Bank (EMDB) and Protein Data Bank (PDB) with accession codes SpRY NAC PAM 20 bp (EMD-40740 and PDB 8SRS), SpRY NGG PAM (EMD-40681 and PDB 8SPQ), SpRY NTC PAM (EMD-40705 and PDB 8SQH), SpRY NAC PAM 0 bp (EMD-41073 and PDB 8T6O), SpRY NAC PAM 2 bp (EMD-41074 and PDB 8T6P), SpRY NAC PAM 3 bp (EMD-41085 and PDB 8T76), SpRY NAC PAM 10 bp (EMD-41079 and PDB 8T6S), SpRY NAC PAM 13 bp (EMD-41080 and PDB 8T6T), SpRY NAC PAM 18 bp (EMD-41083 and PDB 8T6X), SpRY off-target 1 bp (EMD-41084 and PDB 8T6Y), SpRY off-target 6* bp (EMD-41086 and PDB 8T77), SpRY off-target 8* bp (EMD-41087 and PDB 8T78), SpRY off-target 10* bp (EMD-41088 and PDB 8T79), SpRYmer (EMD-41093 and PDB 8T7S), SpG NGG PAM (EMD-41867 and PDB 8U3Y), and SpG NGC PAM (EMD-41775 and PDB 8TZZ). Source data are provided with this paper.

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

## Acknowledgements

We thank I. Stohkendl in the Taylor group for insightful discussions. This work was supported in part by Welch Foundation grants F-1808 (to I.J.F.), and F-1938 (to D.W.T.), the National Institutes of Health R01GM124141 (to I.J.F.), R01AI110577 (to K.A.J.), and R35GM138348 (to D.W.T.), and a Robert J. Kleberg, Jr. and Helen C. Kleberg Foundation Medical Research Grant (to D.W.T.). The content is solely the responsibility of the authors and does not necessarily represent the official views of the National Institutes of Health.

## Author contributions

Conceptualization: J.P.K.B. Investigation and data analysis: G.N.H. (protein production, cryo-EM data collection, structure determination and modeling for SpRY), J.P.K.B. (kinetic analysis of SpRY and structural analysis of SpG) and M.M.H. (biochemistry and kinetic analysis of SpG), K.A.J. and T.L.D. (kinetic analysis), H.Z. and I.J.F. (single-molecule imaging, analysis), Writing initial draft: J.P.K.B. and G.N.H. All authors reviewed, edited, and approved the manuscript. Supervision: J.P.K.B. and D.W.T. Funding acquisition: I.J.F., K.A.J., and D.W.T.

## Competing interests

K.A.J. is the President of KinTek, Corp., which provided the chemical-quench flow and stopped flow instruments and the KinTek Explorer software used in this study. The remaining authors declare no competing interests.
