## [Peer Review File · Nature Communications]

Reviewers' Comments:

Reviewer #1:

Remarks to the Author:

The previous issues have been constructively addressed and the revised manuscript is significantly improved.

Reviewer #2:

Remarks to the Author:

Hibshman et al. have addressed most of my concerns, and the revised manuscript has been improved. I have a few minor points for the authors to consider before publication in *Nature Communications*.

Minor points

Fig. 1: It would be informative to list the mutations in SpRY in the legend of Fig. 1a. The labels for the electrostatic potential bars in Fig. 1d,f (also Fig. 6e) are too small. It might be better to enlarge them or move them to the legends.

Fig. 3: It would be informative to label the spacer nucleotides in the gRNA in Fig. 3c.

Fig. 5d: Seconds -> Time (s)

Extended Data Figs. 1, 4, 7 and 9: It would be better to enlarge the labels or improve the resolution of the images for the cryo-EM statistics.

Extended Data Fig. 3: It would be better to add labels indicating the Cas9 domains.

Extended Data Fig. 9: It would be better to add a schematic showing the Cas9 domains (similar to Fig. 1a) above the structures.

Response to reviewers

Reviewer #1 (Remarks to the Author):

The previous issues have been constructively addressed and the revised manuscript is significantly improved.

We thank the reviewer for their comments and time.

Reviewer #2 (Remarks to the Author):

Hibshman et al. have addressed most of my concerns, and the revised manuscript has been improved. I have a few minor points for the authors to consider before publication in Nature Communications.

We thank the reviewer for their constructive suggestions for the figures in the manuscript. We had incorporated the suggestions as detailed in the response below.

Minor points

Fig. 1: It would be informative to list the mutations in SpRY in the legend of Fig. 1a. The labels for the electrostatic potential bars in Fig. 1d,f (also Fig. 6e) are too small. It might be better to enlarge them or move them to the legends.

We have incorporated the reviewer's alterations. The figure legend of for Fig. 1a now reads:

"Cas9 domain organization with SpRY mutations A61R, L1111R, D1135L, S1136W, G1218K, E1219Q, N1317R, A1322R, R1333P, R1335Q, and T1337R highlighted in light green."

The surface electrostatic potential bars have been enlarged in Fig. 1d,f and Fig. 6e.

Fig. 3: It would be informative to label the spacer nucleotides in the gRNA in Fig. 3c.

The spacer nucleotides have been added to Fig. 3c.

Fig. 5d: Seconds -> Time (s)

The figure has been updated.

Extended Date Figs. 1, 4, 7 and 9: It would be better to enlarge the labels or improve the resolution of the images for the cryo-EM statistics.

The labels in Supplementary Figs. 1, 4, 7, and 9 have been enlarged.

Extended Date Fig. 3: It would be better to add labels indicating the Cas9 domains.

Cas9 domain labels have been added to Supplementary Fig. 3. The figure legend now reads:

“Supplementary Fig. 3. SpRYmer stem loop interaction. **a**, Overview of the SpRYmer atomic model with domains labeled. **b**, Top-down view of the SpRYmer atomic model. **c**, Detailed view of the stem loop kissing interaction.”

Extended Date Fig. 9: It would be better to add a schematic showing the Cas9 domains (similar to Fig. 1a) above the structures.

We believe the reviewer may be referring to Supplementary Fig. 8. A domain schematic has been added to Supplementary Fig. 8.